# Ion Channel Partnerships: Odd and Not-So-Odd Couples Controlling Neuronal Ion Channel Function

**DOI:** 10.3390/ijms23041953

**Published:** 2022-02-10

**Authors:** Nicholas C. Vierra, James S. Trimmer

**Affiliations:** Department of Physiology and Membrane Biology, University of California Davis School of Medicine, Davis, CA 95616, USA; ncvierra@ucdavis.edu

**Keywords:** ion channels, calcium signaling, neurons, protein–protein interaction

## Abstract

The concerted function of the large number of ion channels expressed in excitable cells, including brain neurons, shapes diverse signaling events by controlling the electrical properties of membranes. It has long been recognized that specific groups of ion channels are functionally coupled in mediating ionic fluxes that impact membrane potential, and that these changes in membrane potential impact ion channel gating. Recent studies have identified distinct sets of ion channels that can also physically and functionally associate to regulate the function of either ion channel partner beyond that afforded by changes in membrane potential alone. Here, we review canonical examples of such ion channel partnerships, in which a Ca^2+^ channel is partnered with a Ca^2+^-activated K^+^ channel to provide a dedicated route for efficient coupling of Ca^2+^ influx to K^+^ channel activation. We also highlight examples of non-canonical ion channel partnerships between Ca^2+^ channels and voltage-gated K^+^ channels that are not intrinsically Ca^2+^ sensitive, but whose partnership nonetheless yields enhanced regulation of one or the other ion channel partner. We also discuss how these ion channel partnerships can be shaped by the subcellular compartments in which they are found and provide perspectives on how recent advances in techniques to identify proteins in close proximity to one another in native cells may lead to an expanded knowledge of other ion channel partnerships.

## 1. Introduction

Neuronal voltage-gated ion channels interact with a constellation of proteins that impact their activity, subcellular localization, and associated signaling pathways. Diversity in the gating properties and physiological roles of many voltage-gated ion channels is generated by interactions of the channel’s pore-forming α subunits with auxiliary subunits that control channel activity and membrane trafficking. Voltage-gated ion channels also physically associate with numerous cytoskeletal, anchoring, and scaffolding proteins that play diverse roles in impacting their expression, localization, and function, and that are essential in organizing discrete ion channel signaling systems. The assembly of distinct ion channel signaling complexes and their subcellular distribution are fundamental determinants of neuronal function [1]. In addition to interactions with diverse auxiliary subunits and non-ion channel proteins, some voltage-gated ion channels rely on the partnership with other ion channels to carry out their physiological functions. These ion channel partnerships work to promote efficient coupling of the distinct functions of the partner ion channels to one another and expand the diversity of the physiological events that they underlie.

The indirect coordination of voltage-gated ion channel function through the effects of their ionic currents on membrane potential is well-appreciated. However, close physical association of specific ion channel pairs can facilitate more efficient functional coupling, resulting in more rapid and reliable activation and modulation of their respective functions. Many examples of ion channel partnerships critical for regulating electrical activity are those that work to control and/or quickly respond to elevations in intracellular Ca^2+^. Because Ca^2+^ influx is coupled to diverse signaling pathways that can have profound and sometimes opposing effects on neuronal function [2], and that aberrantly elevated intracellular Ca^2+^ is toxic to cells [3], the sites of Ca^2+^ influx and its intracellular concentration are tightly regulated [4]. Precise spatiotemporal control of Ca^2+^ influx into compartmentalized signaling domains permits rapid and selective activation of Ca^2+^-dependent effectors localized in specialized nanodomains no more than a few hundred nanometers from the Ca^2+^ source [5,6]. The partnership of Ca^2+^-permeable ion channels with Ca^2+^-dependent effectors enables spatially and temporally precise activation of Ca^2+^-dependent processes, including regulation of ion channel function. In neurons, elevations in intracellular Ca^2+^ are generated primarily by the opening of ligand-gated, store-operated, and voltage-gated Ca^2+^ channels (VGCCs) in the plasma membrane (PM), and organellar ion channels localized predominantly in the endoplasmic reticulum (ER) [7].

The molecular composition and subcellular distribution of VGCC-associated signaling complexes is a key mechanism for regulating VGCC-mediated Ca^2+^ signals in neurons [1,8,9]. Early biophysical characterization of VGCCs before their molecular identities were known also led to an alphabetical classification based on their current and gating properties (Table 1). VGCCs are defined at the molecular level by their principal or α1 subunit, which contains the pore-forming and voltage-sensing domains [8]. The 10 genes encoding VGCC α1 subunits in mammals are divided into three phylogenetic subgroups, including the high-voltage activated Cav1 and Cav2 channels, and the low-voltage activated Cav3 channels [8]. All VGCC α1 subunit genes give rise to splice variants that can alter VGCC properties. The α1 subunit of neuronal Cav1 and Cav2 channels assembles with non-conducting auxiliary α2δ and β subunits that influence VGCC membrane trafficking and gating [8].

While VGCCs influence neuronal electrical activity directly through effects on membrane potential via the Ca^2+^ currents they pass, they also control neuronal function by coupling electrical activity to intracellular Ca^2+^ signals that trigger diverse physiological responses. Each of the VGCC subgroups has a characteristic predominant subcellular localization in neurons that defines their distinct contributions to neuronal function [10]. For example, L-type Cav1 channels (LTCCs) are prominently found on the cell body (or soma) and proximal dendrites where they are positioned to mediate Ca^2+^ signals that can rapidly induce changes in gene expression [11,12,13], and also on distal dendrites at or near dendritic spines where they can modulate synaptic function [14,15]. Cav2 channels are a critical component of the synaptic vesicle release machinery and are enriched in presynaptic terminals [16,17]. The physiological properties and functions of Cav3 channels are not as well-studied as those of Cav1 and Cav2 channels, but important roles for Cav3 channels have been identified in regulating electrical excitability in many different types of brain neurons [18]. De novo pathogenic variants in all classes of VGCCs have been linked to debilitating neurological disorders, highlighting the key contributions of these channels to normal brain function [19]. The interactions of VGCC with cytoskeletal, anchoring, and exocytotic machinery proteins are intimately involved in organizing discrete Ca^2+^ signaling systems in different neuronal compartments. For example, the interaction of VGCCs with the synaptic vesicle exocytosis machinery in axon terminals is a prominent and well-explored example of an ion channel–protein partnership that is crucial to brain function [20]. In addition to these interactions, the partnership of VGCCs with numerous other ion channels provides a key mechanism to functionally organize VGCC-dependent Ca^2+^ signaling processes.

## 2. Canonical Ion Channel Partnerships

Functional ion channel partnerships established by a close spatial association of ion channel pairs or higher-order oligomers have been reported for many voltage-gated ion channels. Homomeric ion channel complexes are described for VGCCs, voltage-gated Na^+^ channels, and voltage-gated K^+^ channels (for a detailed review, see [21]). In many such cases, channel oligomerization serves a key role in permitting physical interactions between channels to modulate gating properties, tuning electrical activity. For example, the clustering of Cav1.2 channels promotes physical interactions between adjacent Cav1.2 monomers, enabling their functional coupling and promoting cooperative gating of Cav1.2 channels [22,23]. Below, we describe ion channel partnerships formed between distinct ion channels and detail the key physiological roles linked to these associations.

### 2.1. PM-ER Ion Channel Partnerships

Neuronal Ca^2+^ signals initiated by Ca^2+^ influx through PM VGCCs can be accompanied by ER Ca^2+^ release, amplifying the Ca^2+^ signal beyond that produced by opening of the PM Ca^2+^ channel alone. The storage and release of Ca^2+^ is a fundamental function of the ER, and some regions of the neuronal ER are formed into specialized structures known as subsurface cisternae. Subsurface cisternae come into close contact with the PM (within 20 nm) [24], are prominent in the soma of many brain neurons [25], and in some neurons are enriched with ER Ca^2+^ release channels [26,27]. Ca^2+^ signals arising from Ca^2+^-permeable channels found on the neuronal ER are generated primarily by inositol 1,4,5-trisphosphate receptors (IP_3_Rs) and ryanodine receptors (RyRs). IP_3_R-produced Ca^2+^ signals result from the production of IP_3_ generated by activation of phospholipase C. RyRs are gated by elevations in cytosolic Ca^2+^, releasing ER Ca^2+^ in a process known as Ca^2+^-induced Ca^2+^ release (CICR). RyR-dependent CICR can be triggered by Ca^2+^ entering the cytoplasm through PM-localized Ca^2+^-permeable channels, including VGCCs, especially those localized at PM sites near ER-localized RyRs. RyR-generated Ca^2+^ signals couple to distinct physiological processes in neurons, including regulation of synaptic signaling [28], regulation of gene expression [29], morphological plasticity of dendritic spines in hippocampal pyramidal neurons [30], and the regulation of electrical excitability via activation of PM-localized Ca^2+^-activated K^+^ channels [31,32,33]. While VGCC-dependent activation of neuronal RyRs has been reported for Cav3 channels [34,35], most reports in neurons describing the functional coupling of PM VGCCs with RyRs involve Cav1 and Cav2 channels. We note that, in most cases, the specific isoforms of RyRs (RyRs 1-3) participating in these partnerships have not been defined.

### 2.2. Cav1-RyR

In cardiomyocytes, where the partnership of Cav1.2 channels with RyR2 is well-explored, the opening of Cav1.2 channels localized at junctional dyads, a form of ER–PM junctions, triggers robust CICR [36]. Although the physiological responses to Cav1-RyR-mediated Ca^2+^ signals differ between cardiomyocytes and neurons, the key molecular attributes of Cav1-RyR coupling enabling CICR appear to be shared between neurons and non-neuronal cells. Like cardiomyocytes, membrane potential depolarization increases the frequency of spontaneous Ca^2+^ sparks, or ER Ca^2+^ release events [37], in the soma and proximal dendrites of hippocampal pyramidal neurons [38,39]. Pharmacological blockade of either Cav1 channels or RyRs block these Ca^2+^ sparks, indicating that Ca^2+^ influx through both channel types is required for neuronal Ca^2+^ spark generation [38,39,40].

Although the specific physiological functions of spontaneous Ca^2+^ sparks produced by the neuronal Cav1–RyR partnership are unclear, the functional coupling of these channels requires their juxtaposition at ER–PM junctions. In cardiomyocytes, the distance from the mouth of the Cav1.2 channel and RyRs is around 10 nm [41]. A considerable amount of evidence suggests a similar spatial arrangement for functionally coupled Cav1 channels and RyRs in mammalian brain neurons. Early measurements of VGCC-dependent CICR in the soma of cultured hippocampal neurons using confocal microscopy revealed that this form of Ca^2+^ signaling occurs within 3 µm of the PM [42], and similar observations have been made in cartwheel cells of the cochlear nucleus [33]. Cav1 channel-dependent CICR can be inhibited by ER-resident STIM1 molecules [43,44], indicating that Cav1 channel signaling complexes come into close apposition with the ER [45]. Finally, Vierra and colleagues [40] demonstrated that Cav1 channels co-immunopurify with the voltage-gated K^+^ channel Kv2.1 and RyR-containing protein complexes isolated from chemically crosslinked mouse brain homogenates (Figure 1). These Kv2.1-organized signaling complexes at neuronal ER–PM junctions are responsible for localized spontaneous Ca^2+^ sparks in the soma of hippocampal CA1 pyramidal neurons [40].

### 2.3. VGCC–RyR–K_Ca_

RyR-mediated intracellular Ca^2+^ release couples to a wide range of cellular responses, including activation of Ca^2+^-activated K^+^ (K_Ca_) channels. There exist three families of K_Ca_ channels (large conductance K_Ca_1, small conductance K_Ca_2, and intermediate conductance K_Ca_3) whose members play an essential role modulating neuronal action potential firing patterns [46]. Many examples of RyR–K_Ca_ partnerships in neurons also include VGCCs as a third partner. By opening upon membrane potential depolarization VGCCs provide an action potential stimulus-dependent trigger for CICR leading to activation of K_Ca_ channels, which then contribute to repolarizing the membrane potential to affect subsequent action potential firing.

Members of the small-conductance Ca^2+^-activated K^+^ (SK) channel family are widely expressed in brain neurons where they are important for regulation intrinsic excitability and firing patterns. SK channels are indirectly activated by sub-micromolar elevations in intracellular Ca^2+^ through a requisite physical interaction with calmodulin, which functions as the Ca^2+^ sensor for SK channels (for a detailed review, see [47]). SK channels display heterogeneous and distance-dependent functional coupling to various Ca^2+^-permeable channels in neurons, including VGCCs and RyRs. For example, in cerebellar Purkinje neurons, SK channels are activated by Cav2.1-triggered CICR through RyRs at ER–PM junctions [48]. Double knockout of the junctophilin-3 and -4 proteins, which stabilize ER–PM contacts in these cells, results in impaired electrical activity and motor dysfunction [48].

In contrast to SK channels, the large-conductance Ca^2+^-activated K^+^ channel K_Ca_1.1 (i.e., the BK channel) has a lower affinity for Ca^2+^ and must be localized much closer to the Ca^2+^ source than SK channels to ensure reliable Ca^2+^-dependent activation [49]. Consequently, gating of BK channels is highly dependent on the physical association of BK channels with the Ca^2+^-permeable channels that provide their Ca^2+^ source. Kaufman and colleagues used immunogold electron microscopy to demonstrate that somatic BK channels in cerebellar Purkinje neurons cluster over subsurface cisternae ER–PM junctions enriched in IP_3_Rs [26]. They subsequently showed that, in many types of brain neurons, somatic BK channels exist in PM clusters [50], although the relationship of these clustered BK channels to ER–PM junctions and other intracellular Ca^2+^ sources was not determined. RyR-dependent gating of BK channels serves to promote membrane hyperpolarization, dampening electrical activity and/or shaping action potential firing patterns. As a prominent recent example, in cartwheel interneurons of the dorsal cochlear nucleus, where action potential bursts are a key signaling mechanism, Irie and Trussell [33] found that RyR-mediated Ca^2+^ signals triggered by closely associated Cav2.1 channels activate somatic BK channels (Figure 2B). The Cav2.1–RyR–BK triad provides these neurons with a mechanism to ensure rapid and reliable gating of BK channels to control action potential firing patterns [33]. Indriati and colleagues [51] also reported the colocalization of Cav2.1 and BK channels at somatic subsurface cisternae in cerebellar Purkinje cells, suggesting that the Cav2.1–RyR–BK triad may be present across distinct types of neurons to regulate electrical activity.

### 2.4. VGCC–BK

A close spatial association of BK channels with VGCCs was first inferred from biophysical evidence reported over 30 years ago. Experiments performed in cells loaded with Ca^2+^ chelators, which buffer elevations in intracellular Ca^2+^ and restrict the spatial extent of its action from its source, revealed that some BK channels are tightly associated with intracellular Ca^2+^ sources [52,53]. In addition, the sensitivity of BK currents to pharmacological blockers of different VGCCs can vary according to the subcellular compartment in which they are recorded, indicating the existence of distinct compartment-specific VGCC–BK channel partnerships [54]. These observations led to an effort to identify the specific molecular mechanisms that link BK channels to specific Ca^2+^ sources in neurons.

The functional coupling of VGCCs with BK channels can be reconstituted in heterologous expression systems, suggesting a direct molecular interaction of BK channels with Cav1 and Cav2 VGCCs. In support of this model, mass spectrometry-based analyses of protein complexes immunopurified from mouse and rat brain using BK- and Cav1.2- [55] or Cav2- and Cavβ- [56] specific antibodies revealed that BK channels co-purify with Cav1.2 and Cav2 α1 subunits, VGCC auxiliary subunits, as well as numerous other Ca^2+^-dependent signaling molecules. The results of these proteomics experiments combined with parallel coexpression studies in heterologous cells provided a detailed molecular mechanism explaining how Cav2 channels functionally couple to BK channels in presynaptic terminals [57] where BK currents limit neurotransmitter release (Figure 2C). These studies also revealed that the close spatial proximity of these ion channel partners resulted in a dedicated nanodomain for the Ca^2+^ entering through the VGCCs to trigger activation of the associated BK channels, as shown by the distinct sensitivity of BK channel activation to BAPTA but not EGTA Ca^2+^ chelators [55].

Vivas and colleagues used super-resolution microscopy and immunolabeling to reveal a molecular interaction between Cav1.3 and BK channels in both heterologous cells expressing exogenous channels, and on the somata of cultured neurons, identifying a mechanism whereby BK channels can quickly influence action potential firing through their activation by Cav1.3-generated Ca^2+^ signals (Figure 2D) [58]. In hippocampal CA1 pyramidal neurons, Gutzmann and colleagues found that somatodendritic BK channels physically and functionally couple with Cav2.3 channels to control intrinsic excitability (Figure 2D) [59]. While loss of depolarizing Ca^2+^ currents through Cav2.3 would be predicted to dampen electrical excitability, loss of Cav2.3 channels instead results in increased excitability. This was found to result from reduced BK and SK channel K^+^ currents whose activity depends on Cav2.3-mediated Ca^2+^ influx. These results highlight the powerful impact ion channel partnerships can have on the physiological outputs of individual ion channels, and that these partnerships can lead to functional consequences that are not always predictable based on the physiological role of either individual partner.

Interestingly, in some neurons, BK channels can dynamically couple to distinct Ca^2+^ sources. Whitt and colleagues identified diurnal modulation of BK channels in neurons of the suprachiasmatic nucleus, functionally coupling BK channels to Cav1 channels during the day while favoring their activation by RyRs at night when Cav1 currents are attenuated [60]. These results suggest that regulation of the Ca^2+^ source may be a key mechanism whereby neurons tune the response properties of BK channels engaged in such partnerships. It may be that some VGCC–BK channel partnerships are modulated by additional effectors that transiently affect their functional properties or spatial distribution, for example by dynamically regulating their distance from intracellular Ca^2+^ sources through changes in their PM clustering state.

### 2.5. VGCC–IK

The functional coupling of intermediate-conductance Ca^2+^-activated K^+^ (IK) channels to VGCC-dependent Ca^2+^ entry has also been reported in brain neurons. Like SK channels, IK channels are indirectly gated by intracellular Ca^2+^ via an interaction with calmodulin [61]. They are distinguished from SK channels by their larger K^+^ conductance and distinct pharmacological sensitivities. IK channels were shown to contribute to the slow afterhyperpolarization, a Ca^2+^-dependent K^+^ current that contributes to the regulation of intrinsic excitability and firing patterns [62], although there has been debate over the molecular identity of the channel(s) that generate the slow afterhyperpolarization due to its unique biophysical and pharmacological properties [63]. Sahu and colleagues reported the preferential functional coupling of IK channels with Cav1.3 channels in hippocampal CA1 pyramidal neurons [64]. Spatial and functional coupling of Cav1.3 with IK channels could be reconstituted in heterologous cells when the scaffolding protein densin and Ca^2+^/calmodulin-dependent kinase II (CaMKII) were also present. This same group found that Cav1.3, IK, and RyR2 channels colocalized at somatic ER–PM junctions stabilized by junctophilin-3 and -4 proteins in hippocampal CA1 pyramidal neurons (Figure 3) [65]. Knockdown of these junctophilin proteins impaired generation of the slow afterhyperpolarization, indicating that in CA1 pyramidal neurons, junctophilin-3 and -4 containing ER–PM junctions serve an important role promoting this particular ion channel partnership [65].

## 3. Non-Canonical Ion Channel Partnerships

Over the last decade, several studies have revealed the existence of non-canonical ion channel partnerships. In these cases, two ion channels that do not have an obvious direct relationship in terms of their respective functions form physical and functional associations. As in many of the canonical partnerships summarized above, these non-canonical examples also involve VGCC channels. However, in these cases, the pairings are with voltage-gated Kv channels that are not intrinsically Ca^2+^ sensitive. However, these physical and functional pairings result in regulation of one or the other channel partner to underlie important physiological events.

### 3.1. VGCC–Kv Channel Partnerships

Functional coupling of VGCCs and voltage-gated K^+^ (Kv) channels as mediated by their respective impacts on membrane potential is fundamental to signaling in brain neurons. It has long been appreciated that the regulated cellular coexpression and subcellular colocalization of specific combinations of VGCCs and Kv channels underlie diverse neuronal signaling events. However, there is a growing understanding that physical and functional associations between VGCC and Kv channel partners can serve important roles in neuronal signaling beyond their canonical functions as defined by the respective channels’ separate effects on membrane potential.

Kv channels are highly diverse and like VGCCs, Kv channels are primarily defined by their α subunit, four of which coassemble to form a functional channel. Depending on the Kv channel, the core α subunit tetramer can assemble with Kv subclass-specific cytoplasmic and transmembrane auxiliary subunits, which can dramatically impact Kv channel expression, localization, and function.

#### 3.1.1. Cav3–Kv4

The first example defined in which there exists a physical and functional association between a VGCC and a Kv channel was that between low voltage-activated Cav3 T-type Ca^2+^ channels and the Kv4 K^+^ channels that mediate a major component of transient potassium currents (I_A_) in many brain neurons. In cerebellar stellate cells, this non-canonical pairing of two voltage-gated ion channels leads to enhancement of I_A_ in the subthreshold voltage range by T-type Ca^2+^ channel-mediated Ca^2+^ influx [66]. The authors found that Ca^2+^ entry via Cav3 channels leads to a shift in the voltage-dependence of inactivation of I_A_ towards more positive potentials without affecting activation. This resulted in enhanced I_A_ due to the reduced steady state inactivation of I_A_ which, combined with the lack of prominent effects on activation, results in an increase in the I_A_ window current. Biochemical analyses showed that Cav3.2- and Cav3.3-containing T-type Ca^2+^ channel protein complexes immunopurified from rat brain contained Kv4.2 α subunit and KChIP3 auxiliary subunit polypeptides, known to be component subunits of a prominent population of transient potassium channels found in brain neurons [67]. The KChIP auxiliary subunits of Kv4 channels are members of the EF-hand domain-containing neuronal Ca^2+^ sensor protein family [68], although the role of KChIPs acting as Ca^2+^ sensors to confer physiological Ca^2+^-mediated modulation to native Kv4 channel function in neurons had not been established. The authors were able to reconstitute the T-type Ca^2+^ channel-mediated modulation of Kv4-based I_A_ in experiments in transfected heterologous cells coexpressing these partner ion channels. They then used this system to show that KChIP3 coexpression was required for such modulation. To assess the contribution of KChIP subunits to the T-type Ca^2+^ channel-mediated enhancement of I_A_ in native cells, the authors dialyzed anti-KChIP monoclonal antibodies [69] into stellate cells via a patch pipet. Dialysis of anti-KChIP3 antibodies in particular eliminated the effects of T-type channel-mediated Ca^2+^ entry on inactivation of native neuronal I_A_, mimicking the effects of blocking T-type channels.

This unique coupling of these two voltage-sensitive ion channels was subsequently shown by the same group to underlie the gain in stellate cell inhibitory output in response to reduced extracellular [Ca^2+^], such as what occurs during repetitive climbing fiber stimulation that yields complex spiking in cerebellar Purkinje cells [70]. The authors found that experimentally reducing extracellular [Ca^2+^] led to enhanced steady-state inactivation of the Kv4-based stellate cell I_A_, similar to what is seen upon inhibiting Ca^2+^ entry through Cav3 channels. Similar effects of reducing extracellular [Ca^2+^] in the recording solution were also seen in heterologous cells expressing components of the Cav3–Kv4 complex, with the functional coupling between the VGCC and Kv channel again dependent on coexpression of the KChIP3 auxiliary subunit of the Kv4 channels. Repetitive climbing fiber stimulation leading to complex spike discharge in cerebellar Purkinje cells mimicked the effects of experimentally reducing extracellular [Ca^2+^] on stellate cell Cav3-mediated Ca^2+^ influx and reduced the amplitude of the Kv4-based I_A_. Detailed pharmacological analyses, including the use of the anti-KChIP3 monoclonal antibody, defined the Cav3–Kv4 complex as being the mediator of these responses. The overall effect of reducing extracellular [Ca^2+^] is to reduce the enhancement of stellate cell I_A_ mediated by the Cav3–Kv4 complex (Figure 4A), resulting in increased stellate cell firing rate to sustain inhibitory synaptic input onto Purkinje cells and maintaining cerebellar circuit function during high-frequency stimulation.

#### 3.1.2. Cav2.3–Kv4.2

There are numerous forms of excitatory synaptic potentiation that underlie circuit and behavioral plasticity. Excitatory synaptic potentiation has been well studied in the hippocampal Schaeffer collateral pathway through which CA3 pyramidal neurons provide excitatory input onto CA1 pyramidal cells. One form of experimental postsynaptic excitatory synaptic potentiation in hippocampal CA1 pyramidal neurons is mediated by the partnership of the high voltage-activated Cav2.3 R-type Ca^2+^ channel with the Kv4.2 Kv channel (Figure 2C) [71]. Pharmacological blockade of Cav2.3-mediated Ca^2+^ influx with the tarantula neurotoxin SNX-482 (SNX) led to dynamic boosting of excitatory postsynaptic potentials (EPSPs) in hippocampal CA1 pyramidal neurons [71]. While SNX can also directly inhibit Kv4 channel gating [72], this EPSP boosting was sensitive to the fast Ca^2+^ chelator BAPTA, suggesting that Ca^2+^ entry through SNX-sensitive Cav2.3 channels is coupled to a downstream mediator located in close proximity. This Cav2.3-based pathway was unique from a previously defined signaling pathway whereby Ca^2+^ entry through NMDA receptors activated apamin-sensitive small conductance Ca^2+^-activated K^+^ or SK channels (Figure 2C) to yield dampening of EPSPs [73]. The SNX-induced EPSP boosting was also inhibited by 4-AP used at concentrations where it is a fairly selective blocker of Kv4-containing K^+^ channels. Exogenous expression of a dominant negative isoform of Kv4.2 and dialysis of anti-KChIP monoclonal antibodies through a patch pipet also abolished the SNX-induced boosting of EPSPs. This supported a role for KChIP-containing Kv4.2 channels in coupling the inhibition of Ca^2+^ entry through SNX-sensitive Cav2.3 R-type Ca^2+^ channel activity to EPSP boosting. The authors proposed a model that a Cav2.3–Kv4.2 complex in the dendrites of CA1 pyramidal neurons (Figure 4B) functions in a manner analogous to the Cav3–Kv4 complex in cerebellar stellate cells. In this model, Ca^2+^ entry through Cav2.3 channels enhances Kv4-based I_A_ availability, which then serves to dampen EPSP amplitude.

It should be noted that the authors also defined a distinct mechanism for dynamic regulation of CA1 pyramidal cell EPSP amplitude through the more canonical functional coupling of a Ca^2+^-permeable ion channel, the NMDA-type ionotropic glutamate receptor, to a Ca^2+^-activated SK channel (Figure 2C). It is intriguing that immunogold electron microscopy studies of the localization of these ion channels in dendritic spines of CA1 neurons demonstrate distinct pairwise spatial coupling of the two distinct ion channel pairs that underlie the canonical and non-canonical partnerships. Labeling for both SK channels and NMDA receptors is found within the postsynaptic density [74], whereas that for both Cav2.3 [75] and Kv4.2 [76] is found at sites within spines outside of the postsynaptic density itself (Figure 2C). Subsequent studies employing Cav2.3 knockout mice showed that the effects of SNX to boost the amplitude of CA1 pyramidal cell EPSPs were mediated through Cav2.3 channels [77,78] and not through the reported [72] off-target effects of SNX on the Kv channels mediating I_A_.

While the work of Wang and colleagues indicated that Kv4.2 channels and Cav2.3 channels are in close proximity, Murphy and colleagues recently demonstrated that these two channels physically associate in a protein complex. They identified Cav2.3 channels as molecular partners of Kv4.2 channels by using tandem affinity purification combined with mass spectrometry to isolate and analyze Kv4.2-containing protein complexes from hippocampal neurons [79]. Native Kv4.2 channels co-immunoprecipitated with Cav2.3 in wild-type neurons but not Cav2.3 KO neurons, and imaging experiments using fluorescently tagged Kv4.2 and Cav2.3 channels in hippocampal neurons confirmed that Kv4.2 and Cav2.3 channels form stable complexes in dendrites and at excitatory synapses. Moreover, they determined that Cav2.3 and Kv4.2 likely engage in a direct interaction at a 1:1 channel stoichiometry, enabling Ca^2+^ influx through Cav2.3 to gate the associated Kv4.2–KChIP complex. Acute blockade of Cav2.3 channels using Ni^2+^ reduced I_A_ and decreased surface localization of Kv4.2 channels, suggesting that the Cav2.3–Kv4.2 partnership provides an activity-dependent mechanism to dynamically regulate dendritic excitability. Finally, they demonstrated that the dendritic I_A_ gradient seen in CA1 neurons [80] was altered in neurons from mice lacking Cav2.3 channels, disrupting I_A_ modulation of synaptic currents. These results indicate that the partnership of Kv4.2 and Cav2.3 channels is a key mechanism for regulating synaptic inputs in hippocampal neurons.

The Cav2.3–Kv4.2 partnership seen in glutamatergic CA1 hippocampal pyramidal neurons is not detectable in glutamatergic cerebellar granule cells, which instead have the Cav3–Kv4 partnership previously defined in GABAergic cerebellar stellate cells [81]. Interestingly, there was an anterior–posterior gradient of the amplitude of Cav3 Ca^2+^ current as measured in granule cells in different lobules of the cerebellum, with higher Cav3 Ca^2+^ current amplitudes posterior versus anterior. This resulted in a more prominent functional role for the Cav3–Kv4 partnership in posterior cerebellum. It is intriguing that the Kv4.2 and Kv4.3 α subunits also exhibit an anterior–posterior expression gradient at the mRNA [82], protein [83] and functional [84] level, with Kv4.3 predominating in posterior lobes and Kv4.2 in anterior lobes. This provides a compelling example whereby the patterns of cellular expression of the ion channel partners across the cerebellum can profoundly impact firing in these different populations of granule cells in response to their distinct forms of synaptic input.

### 3.2. Ion Channel Partnerships Involving Non-Conducting Functions of Ion Channels

Voltage-gated ion channel partnerships that utilize the non-conducting functions of a constituent channel such as its scaffolding and/or subcellular targeting properties are not as well-studied as partnerships dependent upon regulation conferred by juxtaposed ionic currents. A prominent example of a partnership that relies upon a non-conducting function of an ion channel is present in skeletal muscle, where the physical interaction of PM Cav1.1 LTCCs with RyR1 channels in the sarcoplasmic reticulum is essential for excitation-contraction coupling. Unlike the coupling of Cav1.2 channels and RyR2 channels in cardiomyocytes, Ca^2+^ conductance through Cav1.1 is not required to trigger RyR-mediated CICR in skeletal muscle [85]. Instead, Cav1.1 channels function as a voltage sensor for RyRs, allowing RyRs to open in response to voltage-induced conformational changes in Cav1.1. Thus, the Cav1.1-RyR1 pairing represents an example of an ion channel partnership that relies on a non-conducting role of one of the channels to impact the function of another through a physical association. In neurons, the partnership of the voltage-gated K^+^ channel Kv2.1 with Cav1.2 and RyRs has emerged as an ion channel partnership dependent upon the non-conducting roles of Kv2.1.

### Cav1.2–Kv2.1

Kv2.1 is expressed at high levels in brain neurons where it is present in large clusters that are prominent on the soma and proximal dendrites, and on the axon initial segment [67]. The sites where Kv2.1 forms clusters in both neurons and in heterologous cells are ER–PM junctions [27,86,87,88,89]. Fox, Tamkun and colleagues showed that not only is Kv2.1 localized to ER–PM junctions, it plays an active role in organizing these membrane contact sites [90]. Subsequent studies have revealed that PM Kv2.1 channels interact with ER VAP proteins to form and maintain these membrane contact sites (Figure 1) [91,92], a nonconducting function of Kv2.1 and its paralog Kv2.2 [93], and mediated by the “PRC domain” [94] present on the cytoplasmic C-terminus of both of these Kv2 α subunits.

Endogenous Kv2.1 channels in cultured hippocampal neurons and in brain neurons are found in close spatial association with clusters of Cav1.2 VGCCs [40]. Studies employing coexpression in heterologous cells revealed that coexpression with a nonconducting mutant of Kv2.1 enhances clustering of Cav1.2 at Kv2.1-containing ER–PM junctions. This results in an increase in LTCC activity and enhanced CICR. The former is a consequence of enhanced cooperative gating of LTCCs upon their coclustering [22], and the latter of the increased spatial and function association of the VGCCs with ER RyRs upon their coclustering with Kv2.1 at ER–PM junctions. Neurons from Kv2.1 KO mice have correspondingly reduced Cav1.2 clustering, and reduced spatial and functional association of LTCCs and RyRs [40].

Studies in vascular smooth muscle cells further supported a role for Kv2.1 in regulating the clustering and activity of LTCCs [95]. O’Dwyer, Santana and colleagues found that myocytes from female mice have larger Cav1.2 clusters, higher intracellular [Ca^2+^], and increased myogenic tone compared to those from male myocytes. These differences paralleled the level of expression of Kv2.1 in female (high) versus male (low) myocytes. Moreover, these sex-specific differences in the clustering and activity of LTCCs were eliminated in myocytes from Kv2.1 KO mice. This study supports a role for Kv2.1-mediated clustering of LTCCs as an important determinant of Ca^2+^ signaling in diverse excitable cell types.

These studies used differences in Kv2.1 expression to gain insights into its role in regulating LTCCs and their coupling to intracellular Ca^2+^ signaling. While not directly addressed experimentally, it is likely that given the role of Kv2.1 in forming and maintaining ER–PM junctions, the overall organization of ER–PM junctions was also impacted in these experimental systems exhibiting differences in Kv2.1 expression. To try to separate the broader role of Kv2.1 in forming ER–PM junctions from its ability to recruit LTCCs to these sites, Vierra and colleagues set out to develop tools to selectively disrupt the latter but not the former [96]. They identified a motif, the “Ca^2+^ channel association domain” or CCAD on the proximal C-terminus of Kv2.1, that was necessary and sufficient for clustering LTCCs at ER–PM junctions. Kv2.1 mutants lacking or mutated within the CCAD were still fully competent to form clusters at and organize ER–PM junctions but they failed to recruit LTCCs to these sites. These characteristics are distinct from mutants in the downstream PRC domain which failed to form ER–PM junctions [91,92,93]. The authors also developed a cell-penetrating interfering peptide based on the CCAD sequence that also eliminated LTCC clustering at Kv2.1-containing ER–PM junctions without impacting these membrane contact sites themselves [96]. The authors used proximity ligation assays [97], which detect protein pairs within 40 nm of one another [98], employing antibodies against Kv2.1 and the Cav1.2 LTCC subunit, to define the homotypic and heterotypic coclustering of Kv2.1 and Cav1.2. They found that these CCAD-based interventions eliminated the spatial association of Cav1.2 with Kv2.1 as well as the Kv2.1-mediated enhanced coclustering of Cav1.2 channels with one another. In addition, these CCAD-based interventions reduced the previously described Kv2.1-mediated enhancement of LTCC activity and CICR in heterologous cells and in cells expressing endogenous channels. These interventions also abolished depolarization-induced and Ca^2+^-dependent activation of the transcription factors CREB and c-Fos in hippocampal pyramidal neurons [96], showing that the Kv2.1–LTCC ion channel partnership plays an important role in supporting the Ca^2+^ signaling that underlies excitation–transcription coupling. The effects of these CCAD-based treatments mimicked and occluded the impact of pharmacologically inhibiting LTCC activity, showing that the Kv2.1-mediated regulation of LTCC activity and its localization at ER–PM junctions was crucial to this form of plasticity. As the association of Kv2.1 with LTCCs is specific to neuronal somata, these results provide further experimental support for the privileged role of somatic LTCCs as supplying the bulk of Ca^2+^ entry leading to depolarization-induced transcript factor activation, as demonstrated in a previous study employing microfluidic chambers to separately activate LTCCs in different neuronal compartments [99]. Together, these studies reveal a unique partnership between a VGCC and a Kv channel mediated through their physical association that impacts the function of the Ca^2+^ channel and its role in downstream Ca^2+^ signaling.

## 4. Restricted Subcellular Compartmentalization of Ion Channel Partnerships

Many ion channels have a highly compartmentalized localization, being found at high densities at specific subcellular sites in neurons [10,67]. As such, the trafficking and retention mechanisms that concentrate ion channels at specific sites in neurons could shape the interactions that underlie selective pairings of specific ion channel partners that underlie the physiologically important partnerships described here. The elevated concentrations of certain candidate ion channel partners at specific sites could be sufficient to provide an environment for selective physical and functional coupling between these ion channels and not others. While the precise subcellular localization of many of the ion channel partnerships detailed above as existing in brain neurons has not been determined, it is clear from biochemical data from brain samples that there exists a direct physical association leading to efficient functional coupling, for example between BK channels and either Cav1-containing VGCCs [55] or Cav2-containing VGCCs [56]. However, there are examples where the partner ion channels have been shown to be selectively co-concentrated at specific sites in brain neurons. In certain cases, we can attribute their colocalization at specific sites to the targeting information present on one of the partners, with the second partner recruited to these sites through this partnership. One example of this scenario is the coclustering of Kv2.1 K^+^ channels and LTCCs at ER–PM junctions on neuronal somata (Figure 1). The primary driver of the coclustering of LTCCs and Kv2.1 at these sites is the PRC motif present on Kv2.1 [94], which through its binding to ER VAP proteins [91,92] not only localizes Kv2.1 to high-density clusters at these sites but also provides the underlying mechanism for forming and/or maintaining the ER–PM junctions themselves [90,93]. Through their interaction with Kv2.1, which occurs through a mechanism requiring the CCAD motif on Kv2.1 [96], LTCCs concentrate at these ER–PM junctions by associating directly or indirectly with Kv2.1 present at these sites. As detailed above, a prominent consequence of this association is to bring the PM LTCCs into close proximity with ER RyRs facilitating CICR [40]. The Kv2.1–LTCC partnership at these sites may also recruit the Ca^2+^ channels to the neighborhood of other Ca^2+^ signaling proteins that are found at these sites [40] and contribute to the privileged role of LTCCs in the Ca^2+^ entry that underlies excitation–transcription coupling. It is intriguing that many of the other ion channel partnerships detailed above also occur at ER–PM junctions, for example the pairing of ER Ca^2+^ release channels and PM BK channels in Purkinje cell somata [26], and the tripartite complex of LTCCs, RyRs and BK channels in cartwheel cells [33].

Another example of a highly localized neuronal ion channel partnership is that of T-type (Cav3) Ca^2+^ channels with KCa3.1 Ca^2+^-activated K^+^ channels at cerebellar Purkinje cell axon nodes of Ranvier [100]. The functional association of Ca^2+^ entry mediated by T-type Ca^2+^ channels to activate KCa3.1 K^+^ channels contributes to action potential repolarization that sustains the excitability mediated by the high densities of voltage-gated Na^+^ channels at these sites. While colocalization of the specific ion channel partners mediating this functional association has not been reported, it is likely the partners are present at the exposed node of Ranvier itself and not in other nodal domains (the paranode and juxtaparanode) where they would be beneath the myelin sheath. It is interesting that BK channels have been found localized to the paranodal region of Purkinje cell axons but not at paranodes on myelinated axons of other brain neurons [101]. Exposure of myelinated Purkinje cell axons to BK channel blockers results in an increase in action potential failure at high firing frequencies [101], suggesting that although they are located under the myelin, BK channels can contribute to axonal function after their activation by an as yet unknown Ca^2+^ source.

A prominent example of ion channel partnerships that are defined by the respective subcellular localization of their partners are those functioning in dendritic spines of hippocampal CA1 neurons (Figure 2C) [71]. Within this already highly restricted subcellular compartment, further compartmentalization of NMDA receptors and KCa2.2 channels [74] in the postsynaptic density (PSD), and Cav2.3 VGCCs [75] and Kv4.2 channels [76,102] at sites in the spine membrane outside of the PSD define their respective functional coupling. While the NMDA receptor contains a PDZ binding motif mediating its localization within the PSD [103], the molecular mechanism restricting KCa2.2 channels to this subdomain, or Cav2.3 and Kv4 channels to sites within the spine but outside of the PSD itself, is not known. Unlike NMDA receptor subunits, KCa2.2 does not contain a PDZ binding motif to direct its localization to the PSD. However, that the PSD localization is selectively seen for the long but not short KCa2.2 splicing isoform [104] suggests that motifs within the domain that distinguishes the two are required for localization to the PSD, and a subsequent study supports that KCa2.2 localization to the PSD involves the scaffolding protein MPP2 [105]. Whether any of the other ion channel partnerships detailed above are located at distinct sites in neurons, and how the selective targeting of candidate ion channel partners to specific membrane compartments defines the partnerships that have been identified in neurons remains an interesting topic for future research.

## 5. Conclusions and Future Perspectives

The canonical ion channel partnerships described here involve a specific Ca^2+^ channel in a physical and functional association with a specific Ca^2+^-activated K^+^ channel. However, there also exist non-canonical ion channel partnerships, such as those between VGCCs and voltage-activated Kv4 K^+^ channels, which are not intrinsically Ca^2+^-activated but onto which KChIP auxiliary subunits confer Ca^2+^-dependent modulation. There can also be non-canonical ion channel partnerships wherein the functional relationships between the partners appear to be solely based on their physical association, as in the example of the LTCC–Kv2.1 partnership. Kv2.1-mediated clustering of LTCCs at ER–PM junctions enhances their clustering and activity and increases their spatial and functional association with ER RyRs. There are not any apparent effects on Kv2.1, whose activity is not intrinsically Ca^2+^-dependent. However, long-term elevation of intracellular [Ca^2+^] can result in calcineurin-dependent dephosphorylation of Kv2.1, impacting its voltage activation properties and eliminating its clustering at ER–PM junctions [106,107]. This could provide the LTCC–Kv2.1 partnership with a homeostatic mechanism to dynamically reverse the Kv2.1-mediated enhancement of LTCC activity and its association with RyRs under conditions when intracellular [Ca^2+^] is elevated. The ion channel partnerships described above typically involve the Ca^2+^ channel partner in an instructive role directing the function of a K^+^ channel, with the roles being reversed in the Kv2.1:LTCC partnership. A direct reciprocal impact of the K^+^ channel on its partner Ca^2+^ channel for the partnerships detailed above have not been described, outside of the effects mediated by changes in membrane potential. Future research may identify bidirectional functional impacts in these partnerships or novel partnerships in which the ion channel partners have reciprocal effects on one another.

It should be noted that while most of the known examples of ion channel partnerships involve Ca^2+^ channels as one of the partners, there likely exist other forms of neuronal ion channel partnerships. For example, Na^+^-activated K^+^ channel activity is coupled to Na^+^ entry through specific Na^+^ channel subtypes but not others in brain neurons [108], in somatosensory neurons in dorsal root ganglia [109], and in uterine smooth muscle [110]. This implies close spatial proximity and perhaps physical association between the Na^+^-activated K^+^ channel and Na^+^ channel partners.

The specific molecular mechanisms that generate and maintain the different partners participating in the physical and functional ion channel partnerships reviewed here have, in most cases, not been defined. In many of the examples above, the partnerships defined in native cells have been reconstituted in heterologous expression systems. This allows for structure–function studies defining the domains and then specific amino acids on the ion channel partners that are necessary and sufficient for generating and maintaining the partnership at the physical and/or functional level. This may lead to insights into the potential impact that pathogenic variants in one or the other ion channel partner can have on the partnership, and that could inform genotype-phenotype analyses. It can also lead to the development of specific tools to experimentally intervene in the physical associations underlying the functional consequences of the partnership in native cells. For the LTCC–Kv2.1 partnership, chimeric and mutant isoforms of Kv2.1 were used to define the CCAD motif mediating Kv2.1 association with the Cav1.2 LTCC, leading to the development of point mutants and a cell-penetrating interfering peptide that were then used to selectively disrupt the Kv2.1:Cav1.2 association in native cells [96]. In addition to defining the molecular features that contribute to stable interactions between ion channels, another important area of future investigation will be to identify mechanisms regulating spatiotemporal relationships between ion channel partners. The examples of circadian modulation of Ca^2+^ source coupling for BK channels in neurons of the suprachiasmatic nucleus [60,111] and neuronal activity-dependent spatial and functional coupling of Kv2.1 and Cav1.2 channels with RyRs in hippocampal CA1 pyramidal neurons [40] demonstrates that ion channel partnerships can be dynamic and reversible. Future studies may reveal molecular mechanisms and spatiotemporal properties of other ion channel partnerships that could lead to further insights into the physiological roles of these partnerships. Understanding these interactions may also provide proof of concept for strategies aimed at disrupting or modulating ion channel partnerships for therapeutic benefit.

Future studies may also reveal other non-canonical pairings, such as the LTCC–Kv2.1 partnership, whose existence could not be predicted on a physiological basis alone. The initial approach that suggested a possible association between Kv2.1 and LTCCs in neurons was an unbiased mass spectrometry-based discovery approach that led to identification of proteins specifically copurifying with Kv2.1 from crosslinked brain samples [40]. Such an approach, which identifies proteins in sufficiently close proximity to be crosslinked to one another, could be used to copurify other ion channels that could be experimentally interrogated for spatial and functional association in heterologous cells and in native cells and tissues. There are numerous proximity labeling approaches [112] that could also be employed, although most of these would require exogenous expression of an ion channel fused to an enzyme capable of mediating the proximity labeling reaction. It is also possible to label cell surface proteins in close proximity to native ion channels using antibody-mediated proximity labeling [113], although to our knowledge, this has not yet been applied to ion channels. Another related approach which has recently yielded important information about the Cav2.3–Kv4.2 partnership is the combination of tandem affinity purification with mass spectrometry [79], which revealed Cav2.3 as a component of Kv4.2-containing protein complexes. Such approaches could lead to identification of other ion channel partnerships, and how they shape physiology above and beyond the respective roles of the individual ion channels and their impact on one another through their effects on membrane potential.

## Figures and Tables

**Figure 1 ijms-23-01953-f001:**
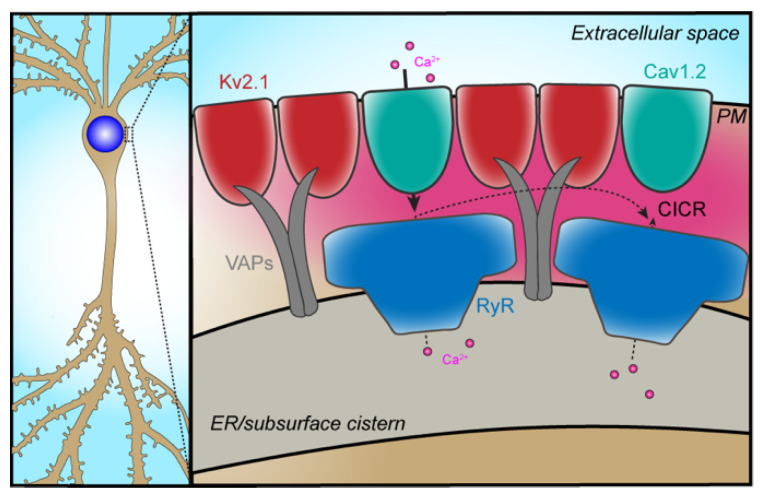
Spatial and functional coupling of Cav1.2 and RyRs at ER–PM contact sites. The left panel is a schematic of a neuron showing that the soma is the subcellular compartment in which this partnership has been defined. The right panel shows that PM Kv2.1 channels (red) form and stabilize ER–PM junctions via a physical interaction with ER-resident VAPA and VAPB proteins (dark grey). Kv2.1 simultaneously promotes the clustering of PM Cav1.2 channels (cyan) at these same sites, placing Cav1.2 channels near RyRs (dark blue) localized in the ER. This arrangement enables Ca^2+^ influx through Cav1.2 channels to trigger enhanced RyR-dependent CICR. Intracellular Ca^2+^ is shown in magenta.

**Figure 2 ijms-23-01953-f002:**
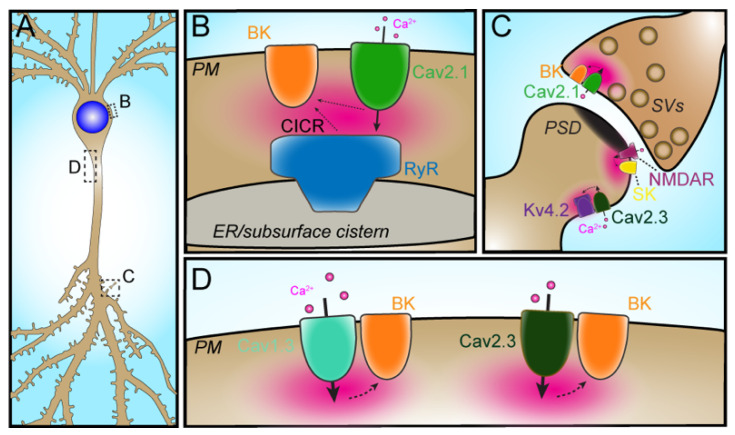
Ca^2+^-activated BK and SK channels form distinct partnerships with different neuronal Ca^2+^ channels. (**A**) Schematic of a neuron showing the subcellular compartments in which the partnerships shown in panels B–D have been defined. (**B**) PM BK channels can partner with ER RyRs and PM VGCCs. In somatodendritic domains of cartwheel neurons, a partnership of PM Cav2.1 channels (green) and ER RyRs (blue) works together to activate spatially and functionally associated PM BK channels (orange). These BK currents work to control patterns of electrical activity in these cells. (**C**) PM BK channels (orange) associated with PM Cav2.1 channels (green) in presynaptic terminals (containing schematized synaptic vesicles [SVs]) are activated by Ca^2+^ influx through these channels. BK current acts to hyperpolarize the membrane potential, limiting Ca^2+^ influx into the terminal by promoting closure of Cav2 channels. In dendritic spines, the partnership of SK channels (yellow) and NMDA receptors (purple) at or near the PSD (schematized in black), and the partnership of Cav2.3 channels (dark green) and Kv4.2 channels (dark purple) at sites outside of the PSD control EPSP amplitude. Together, these partnerships function to modulate distinct aspects of neurotransmission. (**D**) Somatodendritic BK channels (orange) can partner with Cav1.3 channels (light green) and also Cav2.3 channels (dark green). Action potential-triggered Ca^2+^ influx activates BK channels in these partnerships, providing cells with a mechanism to rapidly modulate electrical activity. Intracellular Ca^2+^ is shown in magenta in all panels **B**–**D**.

**Figure 3 ijms-23-01953-f003:**
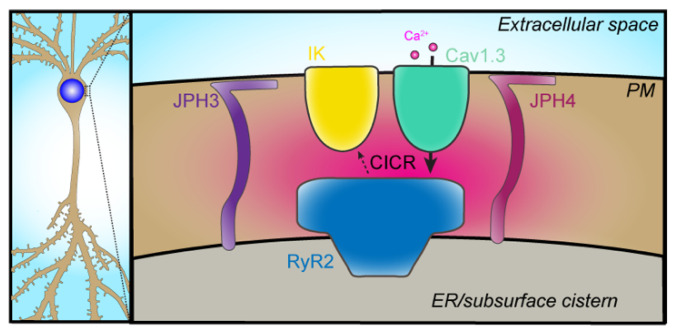
The partnership of somatic Cav1.3, RyR2, and IK channels at junctophilin-3 and -4 stabilized ER–PM junctions. The left panel is a schematic of a neuron showing that the soma is the subcellular compartment in which this partnership has been defined. The right panel shows a schematic of the partnership defined by Sahu and colleagues, who demonstrated that the spatial and functional coupling of PM Cav1.3 channels (green) with ER RyR2 channels (blue) and PM IK channels (yellow) at somatic ER–PM junctions required the presence of junctophilin-3 (JPH3, purple) and -4 (JPH4, magenta) proteins, which form and stabilize contacts between the ER and PM. Loss of JPH3 and JPH4 reduced the spatial association of Cav1.3, RyR2, and IK channels in hippocampal CA1 pyramidal neurons, impairing the CICR-mediated activation of Ca^2+^-activated IK channels and altering intrinsic excitability. Intracellular Ca^2+^ is shown in magenta.

**Figure 4 ijms-23-01953-f004:**
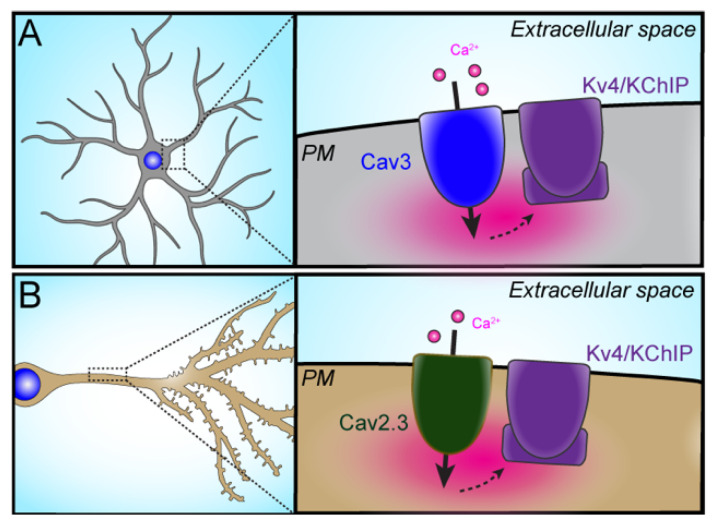
Partnerships of somatodendritic VGCCs with Kv channels. (**A**) The left panel is a schematic of a neuron showing that the somatodendritic compartment is where this partnership has been defined. The right panel shows Ca^2+^ influx through PM T-type Cav3 channels (Cav3 channel in blue) can trigger the activation of associated PM Kv4 channels (purple). (**B**) The left panel is a schematic of a neuron showing that the dendritic compartment is where this partnership has been defined. The right panel shows Ca^2+^ influx through PM R-type Cav2.3 channels (Cav2.3 channel in green) can trigger the activation of associated PM Kv4 channels (purple). In both of these cases, cytoplasmic KChIP auxiliary subunits (shown as purple rectangles) associated with Kv4 channels function as the Ca^2+^ sensor that triggers the opening of Kv4 channels. These partnerships confer Ca^2+^-dependent activation to prominent mediators of I_A_ to dampen electrical excitability. Intracellular Ca^2+^ is shown in magenta in both right panels.

**Table 1 ijms-23-01953-t001:** VGCCs subtypes.

Cav α1 Subunit	Ca^2+^ Current
Cav1.1	L-type
Cav1.2	L-type
Cav1.3	L-type
Cav1.4	L-type
Cav2.1	P/Q-type
Cav2.2	N-type
Cav2.3	R-type
Cav3.1	T-type
Cav3.2	T-type
Cav3.3	T-type

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
