# Peer review of "Ion Channel Partnerships: Odd and Not-So-Odd Couples Controlling Neuronal Ion Channel Function"

_ijms, 2022, doi:10.3390/ijms23041953_

Round 1

Reviewer 1 Report

Nicholas et al. has presented a thorough review on the specific groups of ion channels that couples in controlling neuronal ion channel function. The authors have focused on various partnerships of ion channel like canonical ion channel, non-canonical ion channel and restricted subcellular compartmentalization of ion channel partnerships. Overall, the study is of great interest and for further improvement of the article I have few suggestions or comments to the authors.

Comments:

  • Line 5 Department of is mentioned twice
  • Figure1 indicate the various cellular compartment as it is not clear from the figure about the cellular position of ion channels
  • Figure1also mention in text the specific color used to indicate specific region or channel
  • Figure1 specify specifically which component of VAPs for more clarity
  • Figure2 specify the coloring pattern indicates for which compartment or channel or ion
  • Figure2 keep the labelling pattern same for all panel or figures either mention the channel name outside or inside the channel figures
  • Figure 2 panel (A) Does Cav2 indicates both Cav2.1 and Cav2.2 isoform if yes then change line 221 and mention only Cav2 and write in bracket Cav2.1 and Cav2.2.
  • Figure 2 panel (A) specify what the bubbles in the presynaptic terminal indicates
  • Figure 2 panel (A) is not specified anywhere in the text
  • Figure 2 panel (A) indicates relationship of SK channel and Cav2.3 but in figure legend there is no description for that
  • Figure 2 panel (B) if CICR induce release of Ca+2 from RyR indicate with arrow mark
  • Section 2.2,2.3,2.4 and 2.5 is applicable for all the isoform of RyRs if not then please specify
  • Figure 3 indicate the various cellular compartment as it is not clear from the figure about the cellular position of ion channels. specify the coloring pattern indicates for which compartment or channel or ion.
  • Figure 3 if CICR induce release of Ca+2 from RyR indicate with arrow mark
  • Line 280 mentions RyR2 but figure indicates RyR
  • Rewrite line 357,358,359,360 and 361 according to the figure 4 as figure 4 (A) indicates different the text
  • If figure is included for the section4 restricted subcellular compartmentalization of ion channel partnerships it will be easy for the readers to understand

Reviewer 2 Report

In this article, Nicholas C. Vierra and James S. Trimmer comprehensively reviewed the current progress in Ca2+ channel in a physical and functional association with a specific Ca2+ -activated K+ channel. The ion channel partnerships in neurons is so important for the neuronal functions both in physiological and pathological conditions. The manuscript is well written and organized.

I have one suggestion: whether authors can include some function describe of different ion channel partnership in physiological and pathological conditions. For example, whether the coupling will be changed after disease?

Author Response

We are pleased that Reviewer 2 found that “The manuscript is well written and organized.”  We agree with the reviewer that it is an important research topic to determine how such partnerships are disrupted in disease.  However, as there is no published research on this topic we feel that any discussion of this would be over speculative.

Reviewer 3 Report

The review of Vierra and Trimmer deals with canonical and non channel partnership of calcium ions channel.

The issue is a huge one and different reviews have been published on this subject. Notwithstanding, the paper is well written, structured and with a specific target. The introduction is clear and in the following paragraphs the different features of calcium ion channel described in detail. The topic is afforded taking care of the bibliography. The most recent works are cited as well as the main hypotheses about the non canonical channels partnership. The future perspectives appropriately discussed.

On the whole the paper is of interest to the wide readership of International Journal of Molecular Sciences and deserves publication in the present form.

Author Response

We are pleased that Reviewer 3 found that “On the whole the paper is of interest to the wide readership of International Journal of Molecular Sciences and deserves publication in the present form.”